# RSS Model Improvement Considering Road Conditions for the Application of a Variable Focus Function Camera

**DOI:** 10.3390/s23020592

**Published:** 2023-01-04

**Authors:** Min Joong Kim, Young Min Kim

**Affiliations:** Department of Systems Engineering, Ajou University, Suwon 16499, Republic of Korea

**Keywords:** variable focus function camera, responsibility-sensitive safety (RSS), ISO 21448 (SOTIF), camera sensor, autonomous vehicle, safety

## Abstract

The automobile industry has developed dramatically in recent years, the supply of vehicles has also increased, and thus it has become deeply established in everyday life. Recently, as the supply of vehicles with autonomous driving functions increases, the safety of vehicles is also an emerging issue. Various car-following models for the safe driving of vehicles have long been studied by various people, and recently, a Responsibility-Sensitive Safety (RSS) model has been proposed by Mobileye. However, in existing car-following models or the RSS model, the safe distance between vehicles is presented using only vehicle speed and acceleration information, so there is a limitation in that it cannot respond to changes in road conditions due to the weather. In this paper, in order to ensure safety when the RSS model is applied to a variable focus function camera, an improved RSS model is presented in consideration of the changes in road conditions due to changes in weather, and a safety distance is derived based on the proposed model.

## 1. Introduction

### 1.1. Background

Recently, copious research on autonomous driving has been conducted, and the supply of vehicles equipped wisth autonomous driving functions is increasing [1]. As the supply of vehicles equipped with autonomous driving functions increases, the safety of these vehicles is an emerging problem [2]. The functions of an autonomous vehicle consist of perception, planning, and control, and accidents can occur due to errors in each function, such as perception errors, judgment errors, and control errors [3]. Various vehicle following models such as the Gazis–Herman–Rothey (GHR), safety distance or collision avoidance (CA), and psychophysical or action point (AP) models have been proposed and studied since the 1950s as a way to secure the safety of autonomous vehicles [4]. Today, adaptive cruise control (ACC), which maintains a certain distance from the vehicle in front, is widely used as one of the advanced driver assistance systems (ADAS), which are systems that assist the driver to secure safety while driving [5]. Recently, Mobileye proposed a Responsibility-Sensitive Safety (RSS) model, a mathematical model for determining the fault of an autonomous vehicle [6].

In this paper, in order to secure the safety of autonomous vehicles, car-following models are investigated, and conditions affecting the stopping distance to prevent collisions are analyzed. In addition, in order to secure the safety of autonomous vehicles by applying an RSS model to a variable focus camera, a longitudinal safety distance can be derived from the vehicle follow-up model of autonomous vehicles by responding to changes in road conditions caused by the weather.

### 1.2. Related Work

Brackstone, M. and McDonald, M. (1999) classified the following model into three main categories: the Gazis–Herman–Rothey (GHR) model, the safety distance or collimation (CA) model, and the psychophysical or action point (AP) model and reviewed the time series and evaluation of the classified car-following data [4]. Gipps, P. G. (1981) proposed a new car-following model that reflected the actual behavior of the vehicle and was designed for the parameters of the model to be consistent with the driver’s common sense and reasonable behavior for values of speed and position, and derived safety speeds for the preceding vehicle. By applying variables to the proposed model, traffic flow simulations resulted in mitigation of traffic flow in the time–speed graph [7]. Shalev-Shwartz, S. et al. (2017) presented an RSS model in which autonomous vehicles determined the minimum distance from other vehicles based on five rules to avoid being responsible for accidents, and these rules consider the worst-case scenario for other vehicles and self-driven vehicles during the response time [6]. Chai, C. et al. (2020) applied a modified RSS model by classifying the vehicle-to-vehicle distance into three areas to ensure the safety of autonomous vehicles and to increase efficiency [8].

Xu, X. et al. (2021) identified optimal parameters for increasing the safety performance of RSS models based on SH-NDS data and conducted a study to integrate RSS models with other safety indicators, such as TTC, and predefined driving maneuvers, such as adaptive cruise control (ACC) [9]. Xu, X. (2021) identified that the safety distance presented in the existing RSS model presents a fairly conservative value, an excessively large gap, and presented a new collision avoidance strategy for car-following that balanced safety and efficiency [10]. Liu, S. et al. (2021) extracted interference events from Shanghai Naturalistic Driving Study data and compared human drivers, ACC, and RSS-embedded ACC algorithms for each scenario, confirming that ACC algorithms with RSS models showed better performance than ACC alone algorithms [11]. However, since the SH-NDS data were sampled from a specific region, there was a limit to generalizing driving habits and patterns based on this.

Zhao, C. et al. (2021) proposed an allocation of priority based on inter-vehicle communication as a way to improve lane change performance by utilizing RSS models and confirmed that limited road resources can be utilized through inter-vehicle phenomena [12]. Khayatian, M. (2021) proposed a generalized version of RSS rules applicable to all driving scenarios via vehicle-to-vehicle connections and presented a cooperative driving algorithm for CAV based on the proposed RSS model [13]. Orzechowski, P. F. et al. (2019) suggested a safety verification technique based on the RSS model for merging and intersection scenarios and confirmed that the proposed model guarantees safety and efficiency [14]. In order to apply vehicle-to-vehicle communication or CAV, vehicle-to-vehicle communication (V2V) communication technology for all vehicles must be applied, otherwise there are limitations in its application.

Gutierrez, R. et al. (2021) suggested that self-driving cars have higher reliability than human drivers and have real-time recognition of the surrounding environment to solve traffic problems such as traffic jams and accidents. A scenario was set according to the distance between a vehicle and a pedestrian. In this scenario, the speed of the vehicle was fixed and the time required to stop (or collide) according to changes in lighting and weather (day, night, rain, and fog) was derived [15]. Kordani, A. A. et al. (2018) investigated the coefficient of friction caused by different weather conditions for different pavement surface conditions and the effect of changes in the coefficient of friction on the stopping distance of three types of vehicles: sedans, buses, and trucks [16]. Zhao, G. et al. (2021) considered road friction to evaluate the safety of autonomous vehicles and evaluated the collision probability using the Monte Carlo method by combining pavement friction, reaction time, vehicle spacing, and speed in collision scenarios [17]. Tang, T. et al. (2017) presented a tire-road interaction model to study the effects of various parameters such as axle load, inflation pressure, wheel load, temperature, and water film on the braking distance [18]. Yimer, T. H. et al. (2020) proposed a safe distance vehicle follow-up model describing the safe distance between vehicles and compared it with the 2-s and 3-s rules of safe follow-up distance [19]. In order to secure safety, it is necessary to consider weather conditions and the external environment for the safety of the vehicle during driving, which is suggested through the effect of the change in the friction coefficient of the road surface on the stopping distance.

An improved algorithm based on the Apriori algorithm was proposed for car crash test analysis, and it was claimed that it could represent the relationship affecting crash tests between cars and provide a reference for car design [20]. A method for evaluating passive safety systems was proposed by quantifying energy absorption by analyzing mechanical bump shock absorbers using artificial neural networks (ANN) [21].

### 1.3. Problem Definitions and Paper Composition

Traditional car-following models suggest a safe distance from the preceding vehicle based on speed as a method for safe driving of the vehicle. The recently proposed RSS model also presents a safe distance based on the speed and acceleration of the vehicle. However, in order to avoid collisions with preceding vehicles through the safety distance of the vehicle and to ensure safety of driving, it is necessary to consider various external environments such as the speed of the vehicle as well as the weather environment and road conditions.

As shown in Figure 1, in the research procedure carried out in this paper, various car-following models are investigated, and the safety distance for each model is checked. Subsequently, a method of deriving RSS safety distance considering external environmental factors is presented, and the need to reflect the external environment is proposed when applying the model through the comparison between the safety distance of the existing RSS model and the safety distance reflecting the external environment.

The composition of this paper is as follows. Section 2 introduces measures to secure the safety of autonomous vehicles and various car-following models based on previous studies and describes the limitations of existing car-following models. It also presents the need to consider the external weather environment. Section 3 identifies external weather environment elements and presents an RSS model derived to consider the weather environment. Section 4 discusses the results and their verification, and the last chapter will be a summary.

## 2. Necessity of Research

### 2.1. Existing Car-Following Models

One of the ways to avoid accidents in vehicle operation is to maintain a safe distance from the car in front in order to respond to sudden changes in the preceding vehicles. According to Brackstone, M. and McDonald, M. (1999), classical vehicle follow-up models include GHR, CA, and AP models [4]. Figure 2 shows the basic concept of follow-up theory. When the preceding vehicle decelerates at time t, the following vehicle also decelerates in response after the reaction time elapses.

Gazis–Herman–Rothey (GHR) Model:

As the most widely known model, this model implies that the reaction of the driver of the trailing vehicle is caused by the degree of stimulation felt by the driver, and the main parameters are reaction time and sensitivity coefficient. The basic equation of the GHR model is expressed in Equation (1):(1)$$a_{n}\left(t\right)=\alpha v_{n}^{m}\left(t\right)\frac{\Delta v\left(t-T\right)}{\Delta x^{l}\left(t-T\right)}$$
where $a_{n}\left(t\right)$ represents the acceleration of the nth vehicle at time t seconds, $v_{n}\left(t\right)$ represents the speed at time t seconds, T represents the reaction time, Δv(t) and Δx(t) represents the relative speed and inter-vehicle distance between the nth and (n−1)th vehicles, respectively, and α, m and l are each involved in the speed and distance.

Safety distance or collision avoidance (CA):

This approach dates back to Kometani, E. and Sasaki, T. (1959), and does not describe the stimulus-response type function proposed by the GHR model but manipulates Newton’s fundamental equations of motion to describe safe following distances, as shown in Equation (2) [22]. If the driver follows the following law as shown in Equation (2), a collision occurs.
(2)$$\Delta x\left(t-T\right)=\alpha v_{n-1}^{2}\left(t-T\right)+\beta v_{n}^{2}\left(t\right)+\beta v_{n}\left(t\right)+b_{0}$$

Since advances made by Gipps, P. G. (1981), CA models have been widely used in simulation models.

Psychophysical or action point (AP):

This model was first presented by Michaels, R.M. (1963), and presented the concept that a driver would be able to see that through changes in the vehicle’s apparent size (θ), they are approaching the vehicle ahead [23]. It was suggested that the driver set the value of θ and drove with the limit of the dangerous distance.

The traditional car-following model as described above has been studied in consideration of only the speed of the vehicle, which is a physical value, and the distance between the vehicles. However, driving behavior and safety are greatly affected by external environmental factors such as weather conditions and road surface conditions.

### 2.2. Stopping Sight Distance and RSS Model

Stopping Sight Distance:

The American Association of State Highway and Transportation Officials (AASHTO) presented the Stopping Sight Distance (SSD), as shown in Equation (3). The stopping time is expressed as the distance ($d_{1}$) the vehicle driver has traveled until braking is started by detecting an obstacle ahead and determining the risk possibility, plus the distance ($d_{2}$) the vehicle has moved to come to a full stop after braking.
(3)$$s=d_{1}+d_{2}=0.278vt+\frac{v^{2}}{254\times \left(f\pm G\right)}$$
where s is the stop time period, $d_{1}$ represents the driving distance during the reaction time, and $d_{2}$ represents the braking distance after the reaction time. v represents the driving speed of the vehicle, t represents the reaction time, f represents the coefficient of sliding friction in the longitudinal direction, and G represents the gradient. In this study, assuming the land is flat, the G value was set to 0, and the reaction time, t, was set to 1.7 s, on the results of a comparative study for the driver’s braking response performed by NHTSA.

Responsibility-Sensitive Safety (RSS) Model:

Recently, Mobileye proposed a white-box mathematical model, the Responsibility-Sensitive Safety (RSS) model, to ensure the safety of autonomous vehicles and to be interpreted as a way to determine whether they are negligent [6]. The RSS model presents the minimum safe distance for avoiding collisions in the longitudinal direction and the transverse direction. Figure 3 shows schematic of RSS safe distance for longitudinal direction. Equation (4) represents an RSS safe distance calculation equation for the longitudinal direction. Here, $v_{f}$ and $v_{r}$ denote the speeds of the preceding vehicle ($c_{f}$) and the following vehicle ($c_{r}$), ρ denotes the reaction time, $a_{\max,\;\mathrm{brake}}$ denotes the deceleration of $c_{f}$, $a_{\max,\;\mathrm{accel}}$ and $a_{min,\;brake}$ denote the acceleration during the reaction time of $c_{r}$ and the deceleration until it stops after the reaction time, respectively.
(4)$$d_{\min}^{\mathrm{long}}={\left[v_{r}\rho +\frac{1}{2}a_{\max,\;\mathrm{accel}}\rho^{2}+\frac{{\left(v_{r}+\rho a_{\max,\;\mathrm{accel}}\right)}^{2}}{2a_{min,\;brake}}-\frac{v_{f}^{2}}{2a_{\max,\;\mathrm{brake}}}\right]}_{+}$$
where ${\left[x\right]}_{+}∶=\max\left\{x,\;0\right\}$.

Figure 4 shows schematic of RSS safe distance for lateral direction. Equation (5) shows the calculation formula for RSS safe distance between two parallel vehicles $c_{1}$ and $c_{2}$, where $v_{1}$ and $v_{2}$ are the lateral speeds of $c_{1}$ and $c_{2}$, respectively, and ρ, $a_{\max,\;\mathrm{accel}}^{\mathrm{lat}}$, $a_{min,\;brake}^{\mathrm{lat}}$, and *brake* are the reaction time, lateral acceleration to each other, and lateral deceleration to stop, respectively.
(5)$$d_{\min}^{\mathrm{lat}}=\mu +{\left[\frac{v_{1}+v_{1,\rho }}{2}\rho +\frac{v_{1,\;\rho }^{2}}{2a_{min,\;brake}^{\mathrm{lat}}}-\left(\frac{v_{2}+v_{2,\rho }}{2}\rho -\frac{v_{2,\;\rho }^{2}}{2a_{\;}^{\mathrm{lat}}}\right)\right]}_{+}$$
where $v_{1,\rho }=v_{1}+\rho a_{\max,\;\mathrm{accel}}^{\mathrm{lat}}$, and $v_{2,\rho }=v_{2}-\rho a_{\max,\;\mathrm{accel}}^{\mathrm{lat}}$.

The RSS model also focuses on safety due to the relationship with the driving vehicle, such as securing a safe distance from the preceding vehicle when following the vehicle, and does not consider the sudden risk situation due to changes in the external environment. Therefore, in order to improve the safety and reliability of autonomous vehicles, it is necessary to consider unexpected or dangerous situations that may occur while the vehicle is driving.

### 2.3. Introduction to the Variable Focus Function Camera

As the name suggests, a variable focus function camera is a single camera capable of changing its focus and angle of view. Figure 5 shows a schematic diagram of the concept of a variable focus function camera. The viewing angles of the variable focus camera are 28° (tele), 50° (mid), and 150° (wide) areas, and these cognitive areas are sensor systems that can recognize various ranges by covering the recognition ranges of existing radar and lidar. In addition, the angle of view of the camera may be changed according to the driving speed and environment. For example, when driving at high speed on a highway, a long distance can be measured through a narrow angle, and when driving at low speed, such as in an urban area, a wide area can be sensed through a wide angle.

Additionally, the variable focus camera is a conventional radar sensor or lidar. It can overcome the limitations of sensors while maximizing the advantages of camera sensors and apply surrounding environment adaptive artificial intelligence ISP (image signal processing) technology using weather environment recognition technology through data fusion. For example, there are limitations in which radar sensors do not recognize pedestrians, or situations where multiple pedestrians intersect or are partially obscured by obstacles [24]. LiDAR sensors have a weakness in bad weather such as rain or snow [25]. By using a camera sensor, it is possible to overcome the limitations of radar or lidar sensors, and there is efficiency in terms of space when installed in a vehicle in comparison to using three different cameras depending on the recognition distance. Safety can be secured by maintaining a safe distance by applying an RSS model to a variable focus camera for recognizing vehicles and objects in front.

### 2.4. Need to Consider the External Environment

The National Highway Traffic Safety Administration (NHTSA) categorizes external environmental factors in the hierarchical structure shown in Figure 6. It is largely divided into four subcategories: weather, lighting, particulate matter, and road weather, where weather is again subdivided into rain, temperature, wind, and snow [26].

Recently, ISO 21448 (SOTIF), an international standard for dealing with safety-threatening problems, such as a sensor not recognizing or misrecognizing obstacles due to rapid environmental changes outside or limitations in recognition due to technology or performance, was established. As shown in Figure 7, SOTIFs are classified into four areas, such as known safe scenarios (Area 1), known risk scenarios (Area 2), unknown risk scenarios (Area 3), and unknown but safe scenarios (Area 4), with the end goal being to minimize Area 2 and Area 3 while maximizing Area 1.

Weather conditions, such as rain, snow, and fog, and light effects, such as backlight and shade, degrade the performance of the camera sensor, which may reduce the recognition rate of objects and lanes, or may even lead to them being unrecognized or misrecognized. As a result of this, accidents due to unwanted stops may occur. Table 1 shows the effects of the external environment on image-based recognition systems. By identifying trigger events that cause a dangerous scenario in consideration of the external environment, it is possible to reduce Area 2 and Area 3 areas of the SOTIF, as shown in Figure 7, and secure the driving safety of autonomous vehicles.

In a previous research paper, variables have been identified to apply the RSS model to a variable focus function camera for differential driving vehicles, and RSS models suitable for variable focus function cameras have been derived [27]. However, in the application of variable focus function cameras, one also has to consider the impact of external environments. For the application of SOTIF to variable focus function cameras, external environmental factors were identified and the RSS model was modified to reflect the identified elements in the RSS model to suit the variable focus function camera.

## 3. Methodology

### 3.1. Identification of External Environmental Factors

This paper focuses on the influence of the weather among the external environments in Figure 6. Rain and snow, in particular, affect the cognitive sensor performance, but they are also factors that can change road environmental conditions (e.g., reduce road friction). Road friction further affects road safety, so these factors need to be considered in deriving a safe distance. Therefore, in this study, the friction coefficient of the road surface due to rain and snow was converted into a friction coefficient, and a reference value was set by assuming that the friction coefficient of a clear day was 1 and the friction coefficient was applied by decreasing it in increments of 0.1 to a minimum of 0.2 (considering the minimum value of 0.18 in [17]).

### 3.2. Derivation of RSS Model Using Weather Environments

In the RSS model, the longitudinal safe distance is given in Equation (4); only physical values such as speed and acceleration are considered in deriving the safe distance and there is no friction-related term. Therefore, in this paper, the change in the coefficient of friction was reflected in the term related to deceleration in the longitudinal safety distance derivation equation of the existing RSS model and modified as shown in Equation (6).
(6)$$d_{\min}^{\mathrm{long}}={\left[v_{r}\rho +\frac{1}{2}a_{\max,\;\mathrm{accel}}\rho^{2}+\frac{{\left(v_{r}+\rho a_{\max,\;\mathrm{accel}}\right)}^{2}}{2\left(a_{min,\;brake}\times \mu_{w}/\mu \right)}-\frac{v_{f}^{2}}{2\left(a_{\max,\;\mathrm{brake}}\times \mu_{w}/\mu \right)}\right]}_{+}$$
where μ represents the reference friction coefficient and $\mu_{w}$ represents a friction coefficient that changes according to changes in the weather. The range of $\mu_{w}$ was set to the range of (0.9, 0.8, 0.7, …, 0.3, 0.2), as mentioned in Section 3.1.

## 4. Results and Verification

The safety distance derived from Equation (6) was compared with the stop time of Equation (3) and the existing RSS safety distance of Equation (4), and the vehicle speed was set to 60, 70, 80, 90, 100, 110, and 120 km/h. The response time, ρ (t in Equation (3)), was 1.7 s, which is the result of a comparative study, applied to the driver’s braking response in the Iowa Driving Simulator (IDS) performed by NHTSA [28]. The maximum acceleration and maximum deceleration of the vehicle were set to $4\;m/s^{2}$ and $4.9\;m/s^{2}$, respectively; the values of the full speed range adaptive cruise control (FSRA) system proposed by the ISO 22179 International Standard [29].

Table 2 shows the result of deriving the safety distance for the RSS model including the friction coefficient (Equation (6)) and the SSD (Equation (3)) for each vehicle speed, where the unit of the safety distance is meters. Assuming that the SSD value proposed by AASHTO is the minimum required value, it can be considered safe with regard to collision if the safety distance derived from the proposed RSS model is greater than the SSD value.

Table 2 shows that the SSD stopping distance results for the same speed require a longer safety distance as the friction coefficient decreases, and that a longer safety distance is derived when the friction coefficient is considered according to the road conditions when compared to the existing RSS model (where the friction coefficient is one).

Figure 8 is a diagram showing the result the safety distances according to the change in the speed and friction coefficient for the RSS model modified in consideration of the stop time model and the friction coefficient for Table 2. In the figure, the blue line shows the results of the stop time model, and the red line shows the results of the safe distance of the modified RSS model. It could be verified that the modified RSS model has a larger safety distance at low speeds, whereas the stopping time model has a larger stopping distance at high speeds. This means that the RSS model operates more conservatively at low speeds. It can be seen that the safety distance increases as the coefficient of friction and the speed decrease. Therefore, it was confirmed that the safety distance should account for not only physical quantities such as speed and acceleration, but also for the coefficient of friction on the road surface according to the external weather environment.

The driving scenario was set and verified using MATLAB’s automated driving toolbox. Table 3 shows the initial set value of the simulation. It is a scenario in which the preceding vehicle travels at a constant speed and then stops after a deceleration; the initial distance between the preceding and trailing vehicles is 100 m, the reaction time of the trailing vehicle is 1.7 s, and the friction coefficient of the road surface is set to 0.2.

For the same driving scenario, the speed change of the preceding vehicle was applied as the input value in Equation (4), which is the existing RSS model, and Equation (6), which is the RSS model considering the friction coefficient, to derive the safety distance for each. Vehicle driving control was performed by determining the acceleration/deceleration value of the ego vehicle through the comparison of the derived safety distance and the relative distance between the following and preceding vehicle.

Figure 9 shows the simulation results. Figure 9a shows the change in the position of the preceding vehicle, indicated by the blue line, and the trailing vehicle. When the preceding vehicle stops, it can be seen that the trailing vehicle (indicated by the red line) stops at a certain safe distance. Figure 9b shows the speed change of the preceding vehicle and the trailing vehicle, and when the preceding vehicle finally stops after deceleration, it can be seen that the trailing vehicle also decelerates and stops according to the preceding vehicle. Through this, it can be confirmed that the proposed modified RSS model also secures a safe distance.

Figure 10 shows the results for the SSD, the existing RSS, and the proposed RSS model by applying a friction coefficient of the road surface of 0.2. In the figure, the vertical axis represents the safe distance, the horizontal axis represents the simulation time stamp, the blue line represents the SSD value, the red line represents the result of the existing RSS model, and the yellow line represents the result of the proposed RSS model. While the preceding and trailing vehicles are driving, as shown in Figure 9a, the change in safety distance according to the change in speed as shown in Figure 9b is shown in Figure 10. In other words, the SSD and RSS safety distances are derived by receiving vehicle speed values, so as the vehicle speed decreases as shown in Figure 9b, the derived safety distances gradually decrease, as shown in Figure 10. As illustrated in Figure 10, the safety distance of the proposed RSS model always has a value larger than the SSD result, so it could be deemed safe for longitudinal collisions. However, the RSS model derives conservative safe distances assuming very extreme situations, so while safety can be secured, it is inefficient in traffic flow.

## 5. Conclusions

As the automobile industry continues to develop, the supply of vehicles with autonomous driving functions increases, and vehicle safety is emerging as a distinct problem. Accordingly, various studies have been conducted to derive a safe distance to avoid collisions with the vehicle in front while driving, and recently, an RSS model has been proposed by Mobileye to ensure the safety of self-driving cars and to determine whether they are negligent. In addition, in order to secure the safety of autonomous vehicles, it is necessary to reduce the possibility of accidents caused by unwanted braking due to by the external environment. In this paper, we identify external environmental factors that can affect safety, reflect the identified elements in the existing RSS model, modify the model to suit a variable focus function camera, and derive the safety distance for the modified model. It could be verified that the RSS model presents a more conservative value at low speeds, and a smaller coefficient of friction increases the safety distance value at low speeds. This study confirmed that it is necessary to consider not only physical variables, such as speed and acceleration, but also to consider the effects of the external environment in order to derive a safe distance. In addition, a method for securing safety was proposed by applying different friction coefficient values for deriving the safety distance of the RSS model, suitable for a variable focus camera according to changes in the weather. Future work will study ways to improve traffic flow efficiency as well as ensure safety based on the proposed model.

## Figures and Tables

**Figure 1 sensors-23-00592-f001:**
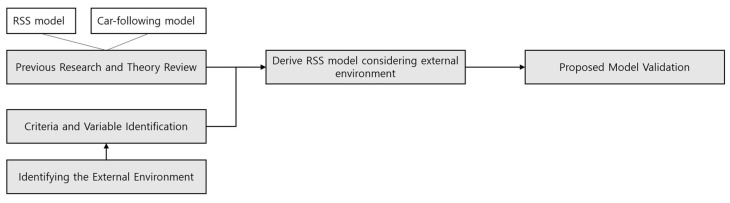
Research scope and objective.

**Figure 2 sensors-23-00592-f002:**
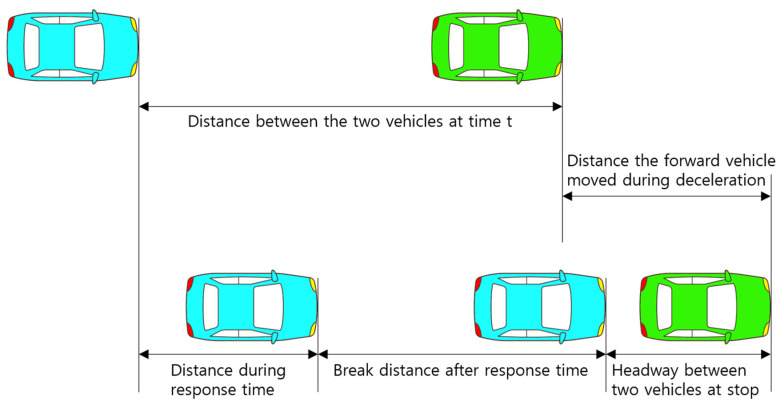
Car-following model concept.

**Figure 3 sensors-23-00592-f003:**
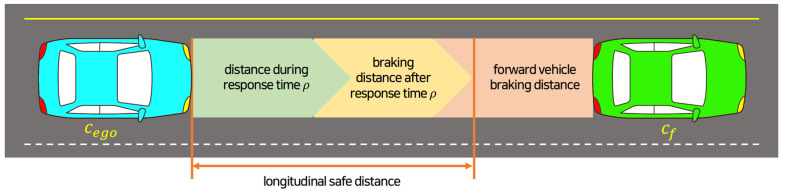
Longitudinal safe distance.

**Figure 4 sensors-23-00592-f004:**
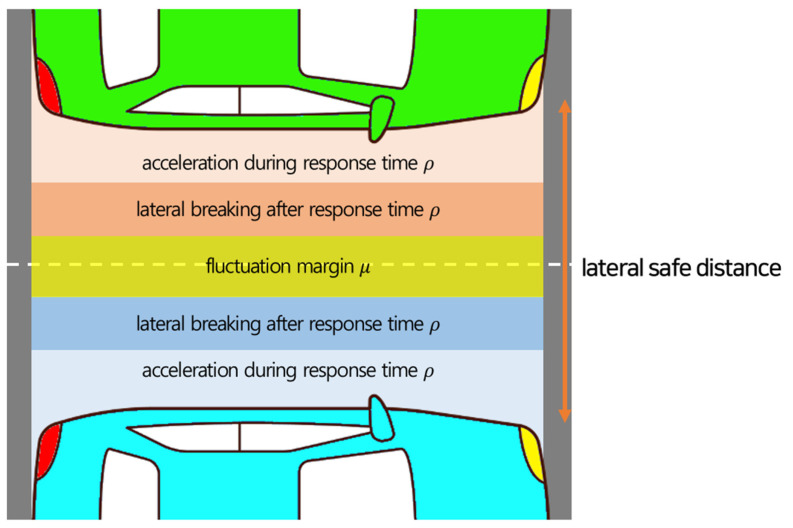
Lateral safe distance.

**Figure 5 sensors-23-00592-f005:**
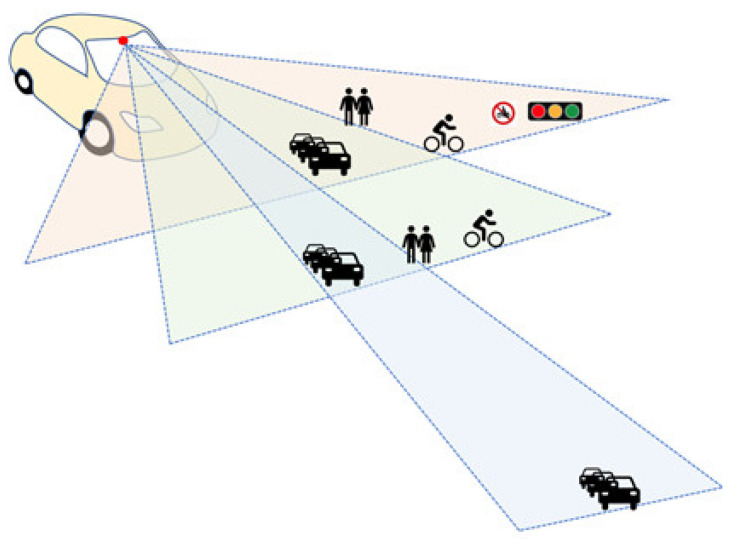
Schematic of a variable focus function camera.

**Figure 6 sensors-23-00592-f006:**
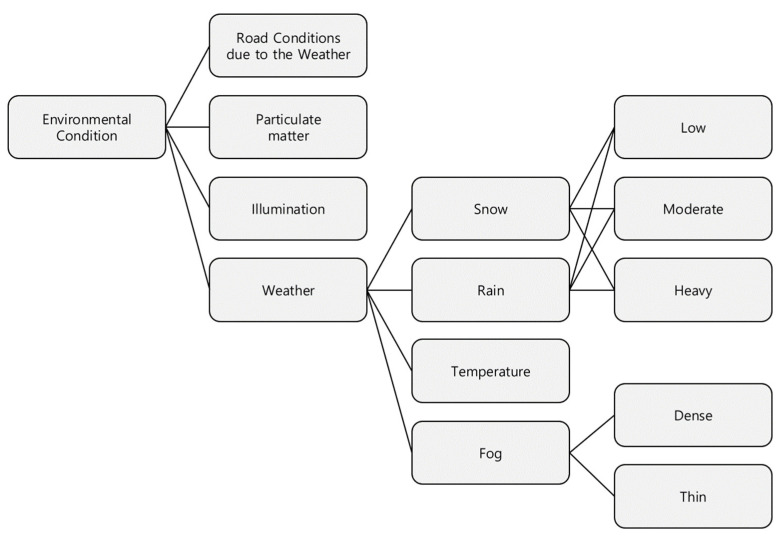
Environmental conditions category.

**Figure 7 sensors-23-00592-f007:**
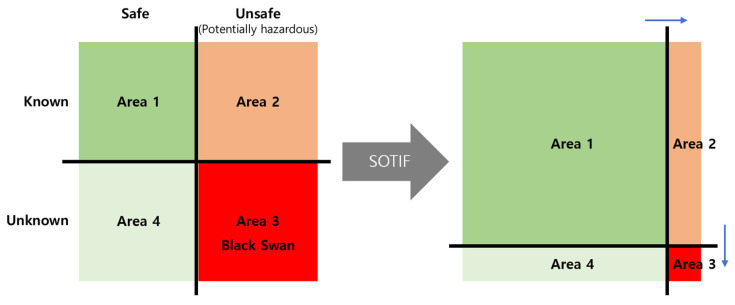
SOTIF Goal.

**Figure 8 sensors-23-00592-f008:**
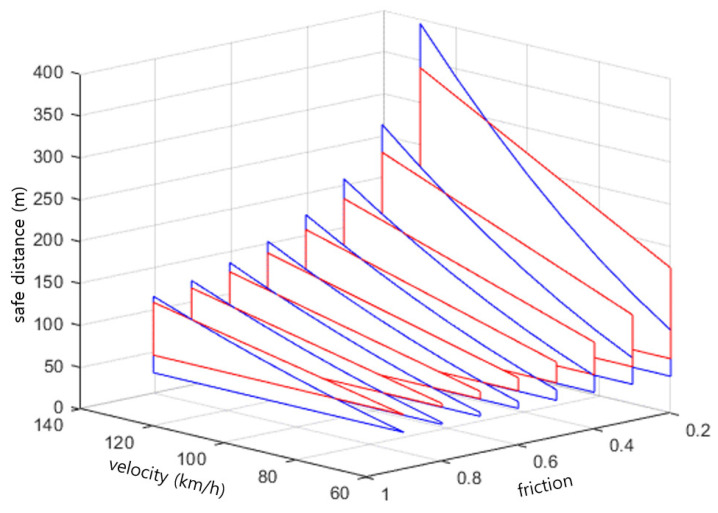
Safety distance according to the change in friction coefficient and speed of each model.

**Figure 9 sensors-23-00592-f009:**
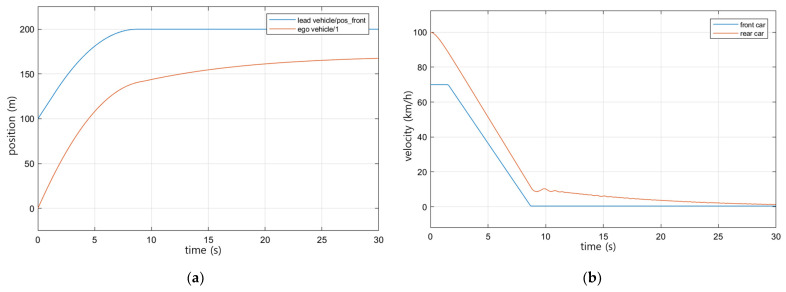
Simulation results: (**a**) changes in the position of the preceding and trailing vehicles; (**b**) speed changes between leading and trailing vehicles.

**Figure 10 sensors-23-00592-f010:**
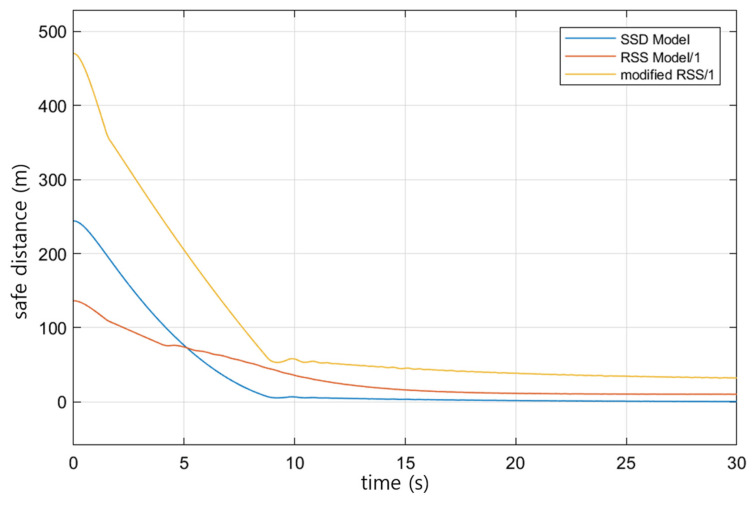
Comparison of safe distances by model. Blue line: SSD model in AASHTO; red line: traditional RSS model; yellow line: proposed modified RSS model.

**Table 1 sensors-23-00592-t001:** Impact on image-based recognition systems by the external environment.

External Environment	Cause	Impact on the Camera Sensor
Luminous source	Backlight	Possibility of failure of recognition or misrecognition by causing photo saturation on the image sensor
	Low luminous	Possibility of the image sensor not recognizing the obstacle due to insufficient amount of light
	difference illumination	Image distortion occurs due to rapid illumination differences, causing the object to not be recognized or misrecognition
Weather environment	Rain	Image distortion caused by water droplets or wipers may cause object or lane recognition or misrecognition
	Snow	Image distortion caused by eye drops or wipers may cause object or lane recognition or misrecognition
	Fog	Visibility constraints caused by fog may cause objects and lanes to be unrecognized or misrecognized
Fine matter	Pollution	Due to contamination by dust, mud, fine matter, etc., obstacles and lanes may not be recognized or misrecognized due to visibility obstruction and restrictions

**Table 2 sensors-23-00592-t002:** Comparative analysis of the safety distance SSD and RSS according to the friction factor.

Safe Distance (m)		Vehicle Velocity (km/h)							
Friction	Model	60	70	80	90	100	110	120	130
1	SSD	42.51	52.35	62.97	74.39	86.59	99.58	113.36	127.92
	RSS	61.96	70.54	79.12	87.69	96.27	104.85	113.42	122.00
0.9	SSD	44.08	54.49	65.77	77.93	90.97	104.88	119.66	135.32
	RSS	65.06	74.06	83.07	92.07	101.08	110.08	119.09	128.09
0.8	SSD	46.05	57.17	69.27	82.36	96.43	111.49	127.53	144.56
	RSS	68.92	78.46	88.00	97.55	107.09	116.63	126.17	135.71
0.7	SSD	48.58	60.61	73.77	88.06	103.47	120.00	137.66	156.44
	RSS	73.90	84.12	94.35	104.58	114.81	125.04	135.27	145.50
0.6	SSD	51.96	65.21	79.77	95.65	112.84	131.34	151.15	172.28
	RSS	80.53	91.67	102.82	113.97	125.11	136.26	147.41	158.56
0.5	SSD	56.68	71.64	88.17	106.28	125.96	147.22	170.05	194.46
	RSS	89.81	102.24	114.67	127.10	139.54	151.97	164.40	176.83
0.4	SSD	63.77	81.28	100.77	122.22	145.65	171.04	198.40	227.73
	RSS	103.73	118.09	132.45	146.81	161.17	175.53	189.89	204.25
0.3	SSD	75.58	97.36	121.77	148.80	178.46	210.74	245.64	283.17
	RSS	126.94	144.51	162.08	179.65	197.23	214.8	232.37	249.94
0.2	SSD	99.20	129.51	163.76	201.95	244.07	290.13	340.13	394.07
	RSS	173.35	197.35	221.34	245.34	269.34	293.33	317.33	341.33

**Table 3 sensors-23-00592-t003:** Simulation parameter settings.

Parameter	Value
Leading vehicle initial speed	70 km/h
Leading vehicle final speed	0 km/h
Following vehicle initial speed	100 km/h
Initial vehicle spacing	100 m
Response time	1.7 s
Friction coefficient	0.2

## Data Availability

Data sharing not applicable.

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
