# Peer review of "RSS Model Improvement Considering Road Conditions for the Application of a Variable Focus Function Camera"

_sensors, 2023, doi:10.3390/s23020592_

Round 1

Reviewer 1 Report

The paper proposes an improved Responsibility Sensitivity Safety model considering changes in road conditions due to weather changes, and a safety distance, in order to improve the safety of vehicles in autonomous driving, however, is necessary to clarify some process used to perform this analysis. Also, some factual clarifications will be helpful as highlighted in the comments below. 

-How can you prevent the algorithm from confusing a state of the road with the objective of the measurement to collision avoidance?

-Is there any effect on instrumentation due to the effects of environment (raining, snow)?

-Is possible to predict an error measurement in seconds including different traffic conditions?

-How is evaluated the passive safety?

-Is possible to include the use of neural networks and machine learning.

-Table 3, the simulation parameters include the use of driver assistance as ABS and so on.

- How these results can be extrapolated to consider other driving conditions (different drivers) instead of controlled conditions?

- It is necessary to include more details of the algorithms and processes used.

-How is validated this proposal?

Please review the next papers and include it in your references.

- Simulation System of Car Crash Test in C-NCAP Analysis Based on an Improved Apriori Algorithm, Physics Procedia 25 ( 2012 ) 2066 – 2071. https://doi.org/10.1016/j.phpro.2012.03.351

-Artificial Neural Networks for Passive Safety Assessment, February 2022 Engineering Letters 30(1):289-297

Author Response

Comments 1: How can you prevent the algorithm from confusing a state of the road with the objective of the measurement to collision avoidance?

Response: You can prevent confusion by defining variable values according to road conditions in advance and then applying those values as input values to the configured collision avoidance model.

Comments 2: Is there any effect on instrumentation due to the effects of environment (raining, snow)?

Response: As shown in Table 1, the environment, such as weather, has an effect on the camera sensor, such as blocking the field of view or distorting the image, causing the object to not be recognized or misrecognition.

Comments 3: Is possible to predict an error measurement in seconds including different traffic conditions?

Response: We expect the time taken from recognition to judgment and control through a variable focus camera to be within 0.2 seconds, the variable focus function camera is a system under development and requires continuous research.

Comments 4: How is evaluated the passive safety?

Response: In this paper, a study was conducted based on active safety by suggesting a safety distance to avoid a collision in a vehicle following scenario.

Comments 5: Is possible to include the use of neural networks and machine learning.

Response: Regarding your suggestions, we will reflect them in future research.

Comments 6: Table 3, the simulation parameters include the use of driver assistance as ABS and so on.

Response: Parameters were set for comparison with existing studies, and additional studies are planned through future studies.

Comments 7: How these results can be extrapolated to consider other driving conditions (different drivers) instead of controlled conditions?

Response: The RSS model can derive a safety distance in response to changes in the opposing vehicle (e.g., deceleration or braking). Therefore, the vehicle can be controlled by deriving the safety distance in the same way for changes in the motion state of the vehicle by other drivers and comparing the derived safety distance with the relative distance.

Comments 8: It is necessary to include more details of the algorithms and processes used.

Response: The content has added to the text. (in red)

Comments 9: How is validated this proposal?

Response: As in Section 4, scenarios were set up and verified using MATLAB's Automated driving toolbox.

Comments 10: Please review the next papers and include it in your references.

- Simulation System of Car Crash Test in C-NCAP Analysis Based on an Improved Apriori Algorithm, Physics Procedia 25 (2012) 2066 – 2071.

https://doi.org/10.1016/j.phpro.2012.03.351

-Artificial Neural Networks for Passive Safety Assessment, February 2022 Engineering Letters 30(1):289-297

Response: The papers have been added to the text and references. (in red)

Reviewer 2 Report

In this paper, an improved RSS model is presented in consideration of changes in road conditions due to weather changes, and a safety distance is derived based on the proposed model. This research is useful to improve the safety of the automated driving systems.

However, the novelty of this paper is relatively low, since it only considers road friction coefficient, and the result is apparently based on easy analysis based the physical movement equations.

The title of the paper mentioned Application of Variable Focus Function Camera, but there is no detailed analysis about it.

The original RSS model is already very conservatively compared to real drivers, which makes the model difficult to be used directly in real automated driving systems, after considering road friction coefficient, make this situation worse.

What’s more, some of the figures need further editing, e.g. Figure 8 and 10 do not have coordinate..

Author Response

Comments 1: The title of the paper mentioned Application of Variable Focus Function Camera, but there is no detailed analysis about it.

Response: In 2.3, the variable focus function camera was introduced, and the contents were supplemented. (in red)

Currently, the variable view angle camera is a system under development, and research has been conducted on ways to secure safety through the application of the RSS model.

Comments 2: The original RSS model is already very conservatively compared to real drivers, which makes the model difficult to be used directly in real automated driving systems, after considering road friction coefficient, make this situation worse.

Response: In this paper, we present the need to consider the road friction coefficient, and we will conduct a study on how to increase efficiency for conservative results through future research.

Comments 3: What’s more, some of the figures need further editing, e.g. Figure 8 and 10 do not have coordinate.

Response: Added axis label to Figures 8 and 10.

Round 2

Reviewer 1 Report

The requested chances have been done.

Author Response

Comments 1: The requested chances have been done.

Response: First of all, we would like to thank the reviewers for their good comments and suggestions for improving the submitted thesis. The points you pointed out were reflected as faithfully as possible in the revised thesis.

Reviewer 2 Report

Figure 8, 9,10 still do not have coordinate axis name, unit.

Author Response

Comments 1: Figure 8, 9,10 still do not have coordinate axis name, unit.

Response: Added axis label units to Figures 8, 9 and 10.
